# PROCEEDINGS A

wave motion, electromagnetism, mathematical modelling

ray tracing, high-frequency wave asymptotics, power balance methods, statistical energy analysis, wireless coverage, channel modelling

**Author for correspondence:**
Gabriele Gradoni
e-mail: gabriele.gradoni@nottingham.ac.uk

One contribution to a special feature 'Innovative and emerging Communications Concepts and Technologies' organized by Ben Allen and Anas Al Rawi

# Wireless power distributions in multi-cavity systems at high frequencies

Farasatul Adnan[1], Valon Blakaj[2], Sendy Phang[3], Thomas M. Antonsen[1], Stephen C. Creagh[2], Gabriele Gradoni[2,3] and Gregor Tanner[2]

[1]Institute for Research in Electronics and Applied Physics, University of Maryland, College Park, MD, USA
[2]School of Mathematical Sciences, and [3]George Green Institute for Electromagnetics Research, University of Nottingham, UK

(ID) SP, 0000-0002-1832-6186; GG, 0000-0001-8321-6883

The next generations of wireless networks will work in frequency bands ranging from sub-6 GHz up to 100 GHz. Radio signal propagation differs here in several critical aspects from the behaviour in the microwave frequencies currently used. With wavelengths in the millimetre range (mmWave), both penetration loss and free-space path loss increase, while specular reflection will dominate over diffraction as an important propagation channel. Thus, current channel model protocols used for the generation of mobile networks and based on statistical parameter distributions obtained from measurements become insufficient due to the lack of deterministic information about the surroundings of the base station and the receiver-devices. These challenges call for new modelling tools for channel modelling which work in the short-wavelength/high-frequency limit and incorporate site-specific details—both indoors and outdoors. Typical high-frequency tools used in this context—besides purely statistical approaches—are based on ray-tracing techniques. Ray-tracing can become challenging when multiple reflections dominate. In this context, mesh-based energy flow methods have become popular in recent years. In this study, we compare the two approaches both in

terms of accuracy and efficiency and benchmark them against traditional power balance methods.

## 1. Introduction

The next generation of wireless communication systems (5G) will be based on super high frequency (SHF) and extremely high frequency (EHF) technologies making use of the enormous amount of bandwidth available at high-frequency bands [1]. Currently, the UK and Europe have adopted carrier frequencies around 3.6 GHz. The Americas and Asia will operate at both sub-6 GHz bands as well as mmWaves bands: more precisely, carrier frequencies at 24 GHz and 28 GHz have been adopted [2,3]. The transition from 4G to 5G and beyond calls for new methods and techniques for modelling network coverage. With a greatly increased density of base stations and demand for massive MIMO [4], beam-steering, peer-to-peer networks and a proliferation of communicating devices in the emerging Internet of Things, accurate modelling of network coverage down to the metre or even centimetre scale is paramount for cost-effective solutions towards maximizing wireless coverage where needed. For recent reviews highlighting the challenges and opportunities for mmWave communication networks, we recommend [2,3]

Traditional, pre-5G channel modelling is largely based on extensive measurement campaigns, collecting data in terms of probability distributions for defined scenarios such as urban, rural, indoor or crowded environments. Important large-scale channel model parameters are free-space path-loss distributions both for line-of-sight (LOS) and non-line-of-sight (NLOS) scenarios, the distribution of the Angle of Departure (AoD) and Arrival (AoA) at the transmitting (Tx) and receiving antennas (Rx), as well as the signal delay spread due to signals taking paths of different geometrical length. Additional parameters are the distribution of different clusters of rays connecting Tx to Rx including multiple reflection and diffraction paths, and various statistical quantities such as auto- and cross-correlations in the large-scale parameters. Such a statistical approach for characterizing channel models has been chosen for the 2007 Winner II (4G) models [5] and extensions thereof [2].

Propagation at mmWave frequencies differs in several important aspects from those at microwave frequencies: free space path-loss is higher, demanding smaller cell sizes (1 km or less) and a higher density of base-stations (BS) together with directive transmission patterns using multiple-input-multiple-output (MIMO) antenna set-ups and beam-forming techniques; penetration loss increases, so shadowing due to buildings and other obstacles has a more detrimental effect, especially in dense urban environments. Diffraction as a transfer path is less efficient with a narrower shadow boundary, so multiple-scattering paths may dominate for NLOS scenarios. The channel characteristics are more sensitive to smaller scales of the environment due to the shorter wavelengths leading to significant, site-dependent diffuse scattering, see [3]. Further complications will arise when considering mobile BSs, device-to-device or vehicle-to-vehicle communication.

A good understanding of the propagation mechanisms and available communication channels is vital for designing and optimizing mmWave networks. There is a growing consensus that purely statistical channel models are no-longer suited, and that one needs to move to (semi) deterministic models containing specific information about the surrounding environment—also referred to as *spatial consistency* by Rappaport *et al.* [3]. The METIS (Mobile and wireless communications Enablers for the Twenty-twenty Information Society) consortium has developed a map-based channel model protocol [6], which includes various deterministic ray-tracing contributions (such as diffractive, specular and diffusive scattering) in a site-specific environment. Establishing a balance between complexity and efficiency is important and is still heavily weighted towards simple models of surrounding obstacles and only a few successive scattering events. Similarly, the MiWEBA (Millimetre-Wave Evolution for Backhaul and Access) approach [7] accounts only for deterministic specular reflections and treats other relevant

quantities statistically. The METIS report [6] importantly states the limitations of ray-tracing: the computational effort of sampling all geometrical paths from a sender to a receiver up to a certain number of reflections $n$ increases quickly with $n$ due to an exponential increase in the number of ray paths. Moreover, the computational demand for a ray-tracing analysis increases with the complexity of the boundaries due to the growing number of possible cases which need to be considered. (Flat walls are easier to handle than sub-structured surfaces, for example). Commercial software for ray-tracing in the mmWave regime considered in [8] gives only fairly coarse resolutions. There is thus considerable doubt that the METIS/MiWEBA approach can be systematically extended and refined beyond the current suggestions.

A possible way to incorporate site-specific information without reverting back to full-wave solvers—far too costly given that typical distances are orders of magnitudes larger than the wavelength—is using full ray-tracing computations. Ray tracing is efficient as long as the number of possible reflections taken into account is small, but may become inefficient in reverberant environments. Recently, a mesh-based ray tracing solver called *Dynamical Energy Analysis* (DEA) [9,10] has been developed. It approximates wave energy transport using energy flow equations written in terms of so-called linear transfer operators, here formulated as a boundary integral equation computing power fluxes through interfaces. DEA provides an efficient numerical approximation of these operators in matrix form and can be formulated on computational meshes, such as provided by finite-element (FE) meshes in two [11] and three dimensions [12]. Recently, a theory has been developed to include the geometrical theory of diffraction in phase space distributions through the Wigner function [13], which paves the way to the inclusion of diffraction in DEA. The advantages of DEA compared to standard ray-tracing are that the complexity of the environment (due to complex boundaries) is fully modelled as part of the mesh and multiple reflections can be accounted for by iterations of the linear operator. The size of the DEA mesh is independent of frequency and allows for large variations in cell size.

In this paper, we analyse the usefulness of the DEA methodology compared to standard ray tracing methods, and simplified power balance (PWB) relations, offering an example of multiply connected and open indoor environments. We will focus here in particular on modelling EM energy distributions in cavities coupled through apertures and including obstacles.

Applications of the PWB method to nested reverberation chambers (RC) to emulate fading in multiply connected wireless channels have been considered in the literature. Both two- and three-cavity RC environments have been studied in the time [14] and frequency [15–18] domains. More recently, hybrid deterministic-statistical models have been developed to study complex single cavity environments [19] in an electromagnetic compatibility (EMC) context and to accelerate ray tracing in wireless coverage planning tools [20]. However, as future wireless communications will seek to improve coverage in the outdoor to indoor transition, there is a need to go beyond single environment modelling, thus consider coupling across multiply connected environments. Recently, a graph model of multi-room environment has been developed [21].

The paper is structured as follows: we overview the different methods mentioned above in a background section, §2; we then define the problem set-up in §3, give a technical introduction of the methods in §4, and present the results in §5. We draw conclusions in §6.

## 2. Background

Determining the electromagnetic (EM) field within systems of enclosures connected through apertures is a standard problem [22] with applications ranging from EMC issues to characterizing wireless communication in indoor scenarios. In principle, one can numerically solve the governing Maxwell's equations using methods such as the finite-element method (FEM) [23], Finite difference time domain (FDTD) method [24] or the transmission-line modelling (TLM) method [25,26]. The resolution required for these deterministic methods scales with the inverse of the wavelength and is thus computationally expensive in the short-wavelength limit (i.e. when wavelengths are small compared to typical scales of the enclosure). In addition, small changes of the interior structure of a given system will drastically alter the solution of the EM field [27,28].

Specific high-frequency and statistical methods may thus be more appropriate when studying such systems describing mean values and fluctuations of the system's response. In the following, we will briefly introduce some of these approaches, namely the *power balance method*, *ray tracing* and the so-called *Dynamical Energy Analysis*. A detailed numerical comparison of these three methods for coupled enclosure scenarios is provided in subsequent sections.

## (a) Power balance methods

Wave energy distributions in complex mechanical systems can often be modelled well by using a thermodynamical approach. Lyon argued as early as 1969 [29] that the flow of wave energy follows the gradient of the energy density just like heat energy flows along the temperature gradient. To simplify the treatment, it is often suggested to partition the full system into subsystems and to assume that each subsystem is internally in 'thermal' equilibrium. Interactions between directly coupled subsystems can then be described in terms of coupling constants determined by the properties of the wave dynamics at the interfaces between subsystems alone. These ideas form the basis of *Statistical Energy Analysis* (SEA) [30] which has become an important tool in mechanical engineering to describe the flow of vibrational energy in complex mechanical structures. Detailed descriptions can be found in text books by Lyon & DeJong [30], Keane & Price [31] and LeBot [32]. In a wider context, and in particular for applications in electromagnetism, this approach is known under the name *PoWer Balance method* (PWB) [33–35]. The method computes the mean power flow between adjacent subsystems assuming—like in SEA—that the power is proportional to the difference in the energy density of the two subsystems. The constant of proportionality (also referred to as the coupling loss factor) depends on the details of the coupling such as the size of the aperture. The energy density in each subsystem is assumed to be constant. This leads to a simple linear system of equations with dimension equal to the number of subsystems. PWB is an energy method and does not incorporate wave effects such as interferences or resonances and can thus not capture the fluctuations inherent in a wave dynamics. A method similar in spirit but including wave effects is the Baum-Liu-Tesche approach [36,37] which analyses a complex system by studying the travelling waves between its sub-volumes. A version of the PWB method estimating the variance of the fluctuations has been presented in [38]. Alternatively, the full probability density of the fluctuations in coupled cavities can be obtained from the *Random Coupling Model* (RCM), where the cavity fluctuations are produced using random matrix ensembles [39,40]. A hybrid PWB/RCM method has recently been proposed in [41].

The PWB is based on two main assumptions: (i) the wave field is uniform in each subsystem as mentioned earlier and (ii) input and output signals within each subsystem are uncorrelated. Assumption (i) is often fulfilled if the coupling between cavities is weak and damping is low. The second assumption is generally fulfilled if the cavity has irregular boundaries (often referred to as being wave chaotic) [42] and the sources are uncorrelated [43]. Whether a given system is well described by a PWB approach is often hard to determine a priori (due to the complexity of the set-up and the number of cavities) and may depend crucially on how the sub-structuring is chosen. A more accurate, but also computationally more demanding high-frequency approach is ray tracing and DEA discussed in the next sections.

## (b) Ray tracing

A method similar in spirit but very different in applications is the so-called *Ray Tracing technique* (RT). The wave intensity distribution at a specific point $r$ is determined here by summing over contributions from all ray paths starting at a source point $r_0$ and reaching the receiver point $r$. It thus takes into account the full flow of ray trajectories. There are many different approaches to set-up efficient RT algorithms; which method to choose depends on the complexity of the reflecting boundaries and the number of reflections which need to be taken into account. We will not go into details here and refer the readers to [44–46]. The methods have found widespread applications in

room acoustics [45] and seismology [47] as well as in determining radio-wave field distributions in wireless communication [48] and for computer imaging software [44].

PWB and RT are in fact complementary in many ways. Ray tracing can handle wave problems well, in which the effective number of reflections at walls or interfaces is relatively small. It gives estimates for the wave energy density with detailed spatial resolution and works for all types of geometries and interfaces. PWB can deal with complex structures carrying wave energy over many sub-elements including potentially a large number of reflections and scattering events, albeit at the cost of reduced resolution. In addition, the quality of PWB predictions may depend on how the subsystems are chosen as well as on the geometry of the subsystems themselves, and error bounds are often hard to obtain.

Ray tracing algorithms, which keep information about the lengths of individual rays, can predict interference effects and thus recreate the fluctuations in a typical wave signal unlike PWB methods. Diffraction effects can be taken into account using Keller's Geometrical Theory of Diffraction [49] or extensions thereof [50].

For a ray tracing treatment in enclosed regions, the trajectories reflect from walls, and refract if the medium is inhomogeneous. When a ray reflects from a wall, energy is deposited in the wall depending on the wall material properties, the angle of incidence and the polarization of the ray. The geometry of the system needs to be specified to implement RT. In cases where the supporting medium is inhomogeneous, spatial variation must be prescribed as well. Since the field is being represented by a discrete set of rays, the solution invariably requires solving for a large number of trajectories to achieve a smooth distribution of the energy density. For the RT results presented later in comparison with DEA and PWB, we use a simple forward algorithm not sampling the ray lengths and thus not including interference. We will also neglect diffraction effects. The RT results will be compared with results from a DEA calculation, the method is briefly described below.

## (c) Dynamical energy analysis

Dynamical energy analysis (DEA) can be interpreted as an Eulerian description of RT. In DEA, the volume is gridded much as it would be for methods to obtain a full-wave solution. However, since resolution on wavelength scales is not required in a ray-tracing simulation, the mesh can be much coarser than would be necessary for solving the underlying wave problem. The quantity of interest in DEA is an energy density, computed as a phase-space flux on the faces of a mesh cell. For example, in a three-dimensional (3D) problem and for monochromatic radiation, one records the power per unit area and per unit solid angle on the mesh boundary of each cell. This quantity is then propagated across the cell and re-tabulated on the facing surfaces. This process is iterated until a steady state is achieved. The propagator itself takes the form of a linear operator which is discretized using basis functions representing phase-space fluxes on the mesh boundaries. Numerical solutions are obtained by solving linear matrix equations of the form

$$(1 - \mathbf{L}) \, \rho = \rho_0, \tag{2.1}$$

where $\rho$ and $\rho_0$ are flux densities representing the equilibrium solution and the source, respectively, and the matrix $\mathbf{L}$ is a finite-dimensional representation of the phase-space propagator. In typical implementations, the dimension of $\mathbf{L}$ is the total number of mesh boundaries [i.e. $3\times$ the number of mesh cells in a triangulated two-dimensional (2D) mesh or $4\times$ the number of mesh cells in a tetrahedral 3D mesh] multiplied by the number of basis functions used to resolve the momentum (or direction) variable.

The method has been introduced first in 2009 [9] and has been refined over the years, mainly for applications in vibro-acoustics. It is now able to run on grids of several million elements in 2D [10,11,51,52] and has been extended to 3D [12,53]. DEA interpolates between PWB and a full RT analysis when increasing the basis size. It thus delivers a refined picture of the energy distribution compared to PWB and by resolving the directionality of the flow, it is insensitive to any sub-structuring condition so important in a PWB approach. By writing the flow equations in matrix form, all reflections are taken into account, so DEA does not suffer from the problem of

exponential proliferation of rays with the reflection number—a limiting factor of RT algorithms in reverberant environments. By mapping the flow equations onto a mesh, the complexity of the geometry of the enclosure is not an issue in DEA—unlike in RT, where finding the ray intersections with boundaries becomes increasingly difficult for more complex boundaries (in terms of shape and number). Setting up the matrix **L** and solving the system (2.1) can of course be time consuming. It is thus of interest to compare the three methods introduced above and better characterize their domain of validity and efficiency. In particular, we expect that (i) DEA and RT agree in the limit of large basis size/large number of rays; (ii) DEA is more efficient than RT in reverberant situations (low damping, enclosures weakly coupled to an outer environment, thus many reflections), whereas RT will be preferable in more open scenarios; (iii) both RT and DEA beat PWB in accuracy at the expense of a higher computational cost, but agree whenever the PWB conditions discussed above are fulfilled.

We note that neither PWB nor DEA carry any phase information; the results obtained give average energy densities and do not try to replicate wave fluctuations due to interference and resonance phenomena. This treatment mirrors the diagonal approximation adopted to obtain the DEA flow equations, where the phase difference between waves disappear under averaging. Interference induced fluctuations between waves can be predicted separately by statistical methods to give a probability distribution around the average energy provided by DEA/PWB discussed in this paper (see [41] for an example involving coupled cavities). Furthermore, it has been shown in [54] that the energy variance can be estimated via the autocorrelation of DEA densities. RT and DEA predictions are accurate in the moderate to high loss regime, which is a reasonable assumption in both indoor and outdoor wireless communication environments.

In the following, we will present for the first time a detailed comparison of the three methods for a set of coupled cavities.

## 3. Problem set-up

We construct a problem of broad interest in coverage planning of wireless communications: channel modelling of indoor propagation of wireless signals and characterization of the outdoor to indoor coupling. This is inspired by multiply connected rooms with objects within them, reminiscent, for example, of floor plans in a building. In modelling the two rooms coupled through windows/doors, we make an analogy with EM cavities coupled by apertures. Furthermore, object geometries are deliberately chosen to be simple. This provides a convenient framework where all the physical mechanisms relevant to signal propagation are present and can be controlled in our simulations: the amount of power leaking from the outdoor through the indoor, as well as the amount of power entering one room from the other, can be scaled by the number and size of apertures in the cavity walls; the losses experienced by the signal reflected within the environment can be controlled by an average wall absorption factor that will be defined later on; position and numerosity of objects can create or suppress the LOS between pairs of aperture in the interior regions.

The geometry of the problem for which we compare results is shown in figure 1. It is a 2D, two-cavity system, where the cavities are separated by an aperture. There are two ports, one on the top boundary of each of the left and right cavities. These ports are treated as apertures and allow for both the entrance and exit of power from the system. Within each cavity there are scatterers that are shown as circles. In figure 1a, the three-disc configuration of cavity 1 can easily keep rays entering Port 1 within the cavity for a long time before bouncing off to cavity 2. In figure 1b, the three-disc configuration of cavity 1 easily allows rays entering through Port 1 to get inside cavity 2 by bouncing off the intermediate sphere. This will be shown explicitly in the next section. We assume that the boundary of the cavity and the scatterers are made of some material, whose loss is captured by a lossy reflection coefficient—related to the loss factor $\alpha$ of the cavity. We further assume that scatterers and walls are made of the same material, thus leading to the same value of the reflection coefficient throughout. However, the code can accommodate objects and walls with unequal and inhomogeneous reflection coefficients. We have, here, treated

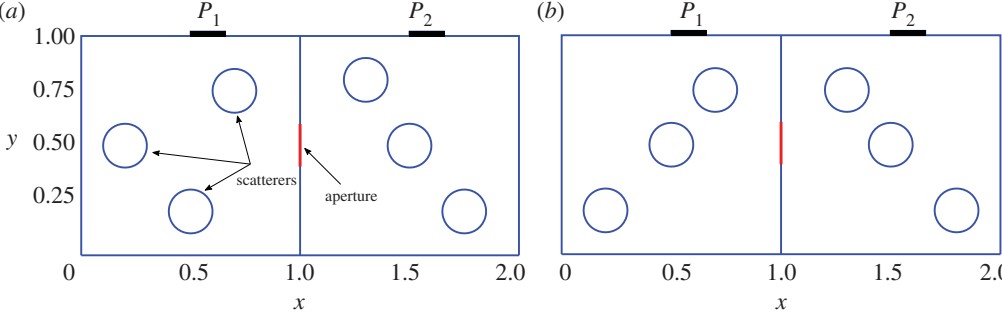

**Figure 1.** Geometries considered for the comparison between the three high-frequency methods. In both the asymmetric (*a*) and symmetric (*b*) configurations, two cavities of unit side-length are connected through an aperture of 0.2 unit length. Each cavity is equipped with a port connected to an external environment which is considered as an infinite-acting reservoir. The embedded circles are made of the same partially reflected material as the wall and can be considered as scatterers. All circles have the same radius of 0.1 units. Both ports have an opening size of 0.1571 unit length. (Online version in colour.)

**Table 1.** Dimensions (in unit length) of the ports (apertures) for the geometries considered in this paper.

| # port | figure 1*a* | figure 1*b* | figure 2 | figure 3 |
|---|---|---|---|---|
| $P_1$ | 0.1571 | 0.1571 | 0.1571 | 0.01571 |
| $P_A$ | 0.2 | 0.2 | 0.2 | 0.02 |
| $P_2$ | 0.1571 | 0.1571 | N/A | N/A |

the loss in an approximate way by making it independent of angle of incidence. However, it is straightforward to account for polarization and include the Fresnel reflection coefficient in the formulation, as usually done in standard RT algorithms. We inject wave energy in Port 1. Then the waves propagate through the cavities and reflect from the walls and the scatterers. After some cavity dwell time, the energy will leave the system either through Port 1, Port 2, or be absorbed by a wall. Our goal is to calculate how much power is delivered either to Port 1 or 2 while we vary the loss of wall and object boundaries. Table 1 reports the dimensions of port 1 ($P_1$) in cavity 1, port 2 ($P_2$) in cavity 2, and the aperture ($P_A$) in the wall connecting the two cavities.

## 4. Calculation of power transfer

We will compare the predictions of RT and the DEA to those of the PWB. To quantify the energy flow predicted by these methods, we define energy fluxes across apertures or ports connecting the two cavities and connecting each cavity to the exterior.

### (a) Power transfer in the power balance method

We begin by defining the quantities used to implement the power balance method. The widths of Ports 1 and 2, which connect the cavities to the exterior, and of the aperture that couples the two cavities are respectively denoted ($w_1, w_2, w_A$). The perimeters of Cavities 1 and 2, including the scatterers, are denoted ($l_1, l_2$). Then,

$$\sigma_i^{\text{wall}} = \alpha(l_i - (w_i + w_A)), \tag{4.1}$$

where $i = 1, 2$ labels the cavities, defines an effective absorption cross sections of cavity $i$, which captures the power absorbed by the wall and scatterer boundaries. The quantity $\alpha$ is the local fraction of incident power absorbed by the boundary. In our 2D system, the absorption cross

section in (4.1) represents an effective physical length. The power absorbed is proportional to the product of this length and the energy per unit area in the cavity under consideration. The constant of proportionality scales with the wave speed. In addition, the power lost by escaping through the cavity walls is proportional to the fraction of power incident $\alpha$, which in practice is determined by the electrical characteristics of the wall material. Here, we have not focused on specific/prescribed wall materials: We rather performed a study by varying $\alpha$ across the full range, from 0 (the incident power is entirely reflected) to 1 (the incident power is entirely absorbed).

The total loss for each cavity, including radiation through apertures and ports, is then

$$\sigma_i^{\text{tot}} = \sigma_i^{\text{wall}} + w_i + w_A. \tag{4.2}$$

Similarly to $\sigma_i^{\text{wall}}$, $\sigma_i^{\text{tot}}$ in (4.2) represents the total length through which the power escapes the cavity. Let

$$P_i^{\text{tot}} = P_i^{\text{inj}} + P_{ij}^{\text{back}} \tag{4.3}$$

denote the total power entering Cavity $i$, which includes both the power $P_i^{\text{inj}}$ directly injected into the cavity and the power $P_{ij}^{\text{back}}$ passing through the aperture from the other cavity, labelled $j$. In detailed calculations to follow, we inject power only into Cavity 1, so $P_2^{\text{inj}} = 0$. We also denote by $P_i^{\text{port}}$ the power leaving through Port $i$ and by $P_i^{\text{wall}}$ the power absorbed by the walls of Cavity $i$ (including those of the scatterers).

Under power balance assumptions, the total power and the powers leaving Cavity $i$ through Port $i$, through the aperture to the other cavity, and being absorbed by its walls are related by

$$\frac{P_i^{\text{tot}}}{\sigma_i^{\text{tot}}} = \frac{P_i^{\text{port}}}{w_i} = \frac{P_{ji}^{\text{back}}}{w_A} = \frac{P_i^{\text{wall}}}{\sigma_i^{\text{wall}}}, \tag{4.4}$$

where $j = 1$ if $i = 2$ and $j = 2$ if $i = 1$. The weighted powers in these equalities provide a coarse-grained analogue of the flux density $\rho$ that is central to DEA. They are equal if $\rho$ is a constant, which amounts to an assumption of ergodicity and low loss. If $\rho$ deviates strongly from uniformity, the power balance assumptions fail.

In the special case $P_2^{\text{inj}} = 0$, we can show from these balance conditions that the total power entering Cavity 1 is

$$P_1^{\text{tot}} = \frac{P_1^{\text{inj}}}{1 - w_A^2/\sigma_1^{\text{tot}}\sigma_2^{\text{tot}}}. \tag{4.5}$$

We can then also easily find the fractions of power lost by each of the mechanisms of wall loss or radiation through ports and apertures. For example, in the absence of wall loss ($\alpha = 0$) the fraction of injected power leaving Cavity 1 through Port 1 is

$$\frac{P_1^{\text{port}}}{P_1^{\text{inj}}} = \frac{w_1(w_2 + w_A)}{w_1 w_2 + w_A(w_1 + w_2)}. \tag{4.6}$$

For the quoted parameters, we find $P_1^{\text{port}}/P_1^{\text{inj}} = 0.641$. On the other hand, with maximum wall loss ($\alpha = 1$), we find

$$\frac{P_1^{\text{port}}}{P_1^{\text{inj}}} = \frac{w_1/\sigma_1^{\text{tot}}}{1 - w_A^2/(\sigma_1^{\text{tot}}\sigma_2^{\text{tot}})}. \tag{4.7}$$

For the quoted parameters, $\sigma_1^{\text{tot}} = 5.995$, and thus $P_1^{\text{port}}/P_1^{\text{inj}} = 0.0267$. Note that in the high loss case the fraction of power leaving through Port 1 is a finite number. We will find in the next section, that in the other two methods, which treat the propagation of wave energy, the power escaping through the ports vanishes in the high loss case. This is because the propagating wave energy invariably encounters a wall and is absorbed before arriving at a Port. However, at low and intermediate loss levels the PWB formulae will be found to be quite accurate.

## (b) Power transfer in the RT method

We briefly discuss the RT algorithm used to calculate power delivery in this section. We basically follow one of the approaches discussed in [55]. Specifically, we use the 'Reflected and Transmitted Rays' approach as discussed in [55, Section III. B] and the power calculation is done using the 'Shooting and Bouncing Ray (SBR) Method' as discussed in ([55], Section IV. C). At a port we launch rays normal to the boundary and uniformly distributed over the port. Each ray contains initially the same amount of power. The rays follow straight trajectories until they encounter a wall or scatterer at which point they are specularly reflected and their power is reduced according to

$$P_{n+1} = |R|^2 \, P_n, \tag{4.8}$$

where $n$ refers to the bounce number, $P_n$ is the power contained in the ray after the $n$-th bounce and $R$ is the power reflectivity (discussed further in the next section). So, after each bounce the ray loses $(P_n - P_{n+1})$ amount of power. The $R$ in this paper is independent of incident angle of the rays and we can fix a value of $R$ for a particular run. Rays that encounter the aperture are allowed to pass straight through, and rays that encounter a port are allowed to leave. We add up all the ray powers when the rays escape through a port and that gives us the total power delivered to a port. To monitor the energy in each cavity, we sum the contributions from each ray segment in a cavity, taking the product of the power and the time of flight of the segment. Numerical details are as follows. We typically launch 8002 rays from port and we let the rays bounce until they escape through either port 1 or port 2 (as shown in figure 4b). We have found 20 bounces to be enough for all rays to escape.

## (c) Power transfer in DEA

In order to quantitatively compare DEA, RT and PWB, we calculate the energy flux across apertures. In the DEA approach, one obtains such a flux by sampling the phase-space density $\rho$, obtained by solving (2.1), along an aperture or port labelled $X$, according to

$$P_X = \langle \rho, \chi_X \rangle, \tag{4.9}$$

where $\chi_X$ is a state vector that is entirely supported on the mesh faces forming the aperture or port. The inner product $\langle, \rangle$ here is defined by

$$\langle \mu, \chi \rangle = \sum_{\text{mesh faces}} \int_{\text{mesh face}} \mu^*(s,p) \lambda(s,p) \, \mathrm{d}s \mathrm{d}p,$$

where $(s,p)$ denote position and momentum coordinates on each face [51]. By choosing $\chi_X$ accordingly, $P_X$ provides a direct analogue of the powers $P_i^{\text{port}}$ and $P_{ij}^{\text{back}}$ used in §4a to characterize the PWB.

A benefit of the DEA approach is that it provides us with a direct means of mapping how strongly different parts of the cavity or its phase space couple to particular loss mechanisms such as transmission through a given aperture. To see this, let us alternatively express the power flux across aperture $X$ in the form

$$P_X = \langle \rho_0, \mu_X \rangle, \tag{4.10}$$

where

$$\mu_X = \left( \frac{1}{1 - \mathbf{L}} \right)^T \chi_X \tag{4.11}$$

is a density, independent of the source $\rho_0$, that quantifies how strongly each element of phase space couples to aperture $X$. Here, $T$ denotes a transpose operation. Note that $\mu_X$ is found in practice by solving an adjoint analogue of (2.1), but driven by the aperture state $\chi_X$ rather than the source $\rho_0$.

An example of such a calculation is shown in figure 2, where the adjoint density $\mu_X$ has been computed for each of the Port and aperture of Cavity 1 in the geometry illustrated in

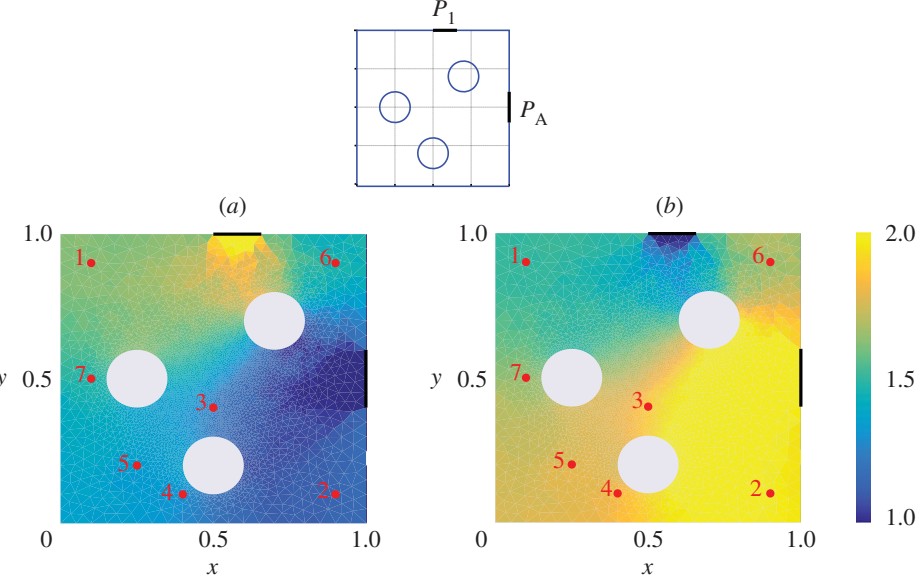

**Figure 2.** We show heatmaps that quantify how strongly different parts of Cavity 1 couple to apertures and ports for the shown geometry (top) and with no wall losses ($\alpha = 0$). These are obtained by integrating adjoint density $\mu_X$ over momentum coordinates. Part (a) shows coupling to the top aperture while part (b) shows coupling the Port on the side section. For reference, red dots correspond to the location of point sources shown in table 2. (Online version in colour.)

**Table 2.** Location of point sources considered for the comparison between DEA, RT and PWB.

| # source | 1 | 2 | 3 | 4 | 5 | 6 | 7 |
|---|---|---|---|---|---|---|---|
| $(x, y)$ | (0.1, 0.9) | (0.9, 0.1) | (0.5, 0.4) | (0.4, 0.1) | (0.25, 0.2) | (0.9, 0.9) | (0.1, 0.5) |

figure 1a, and with no wall losses ($\alpha = 0$). For the visualization in figure 2, we have integrated over momentum coordinates to produce a density in spatial coordinates, plotted as a heatmap. The points labelled 1...7 indicate the locations of sources reported in table 2, which are to be used in the next section to compare predictions of PWB, RT and DEA. The non-uniformity of this heatmap offers a quantitative insight into deviation from the assumptions of ergodicty behind the power balance method: the relatively large aperture losses in this example result in a deviation from the uniformity assumed by PWB that is substantial. By contrast, a second calculation for a set-up that is identical except that the apertures and ports are much narrower, shown in figure 3, results in much more homogeneous coupling. Here, the assumptions of power balance are much better grounded and we expect better agreement between PWB, RT and DEA.

## 5. Results and discussion

In figure 4a, we show the computed spatial distribution of wave energy density using DEA for the geometry of figure 1b, and in figure 4b we show a sample RT trajectory for the geometry of figure 1b. The incident power is launched normal to the boundary through Port 1. In the DEA calculation shown in figure 4a, the colour scale indicates the level of wave energy density. The energy density peaks near the source, that is, Port 1, and decays as one approaches the aperture and passes into cavity 2. In particular, it reaches its minimum in the regions close to Port 2. In the corresponding RT calculation, a total of 8002 rays were launched normal to the boundary of Port 1, and each ray is followed for 25 bounces off the wall and object boundaries. The plot 4b shows

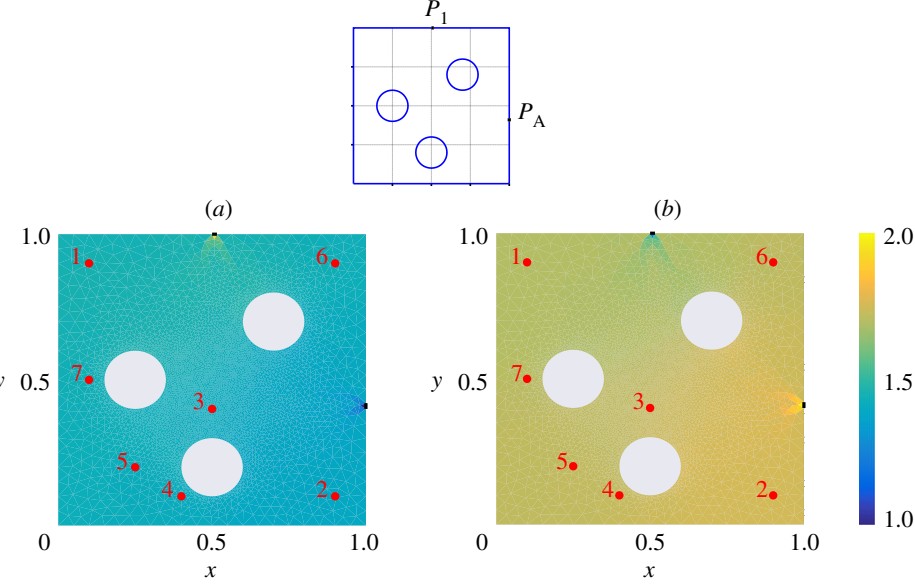

**Figure 3.** We show heatmaps equivalent to those of figure 2 except that the Port and aperture are narrower (reduced by a factor of 10 compared to figure 2). The resulting smaller loss in the dynamics of each cavity results in an adjoint density that is significantly more uniform than in figure 2: significant non-uniformity is seen only in the vicinity of the Port and aperture themselves. This geometry therefore provides a setting that is closer to the idealized assumptions of PWB. (Online version in colour.)

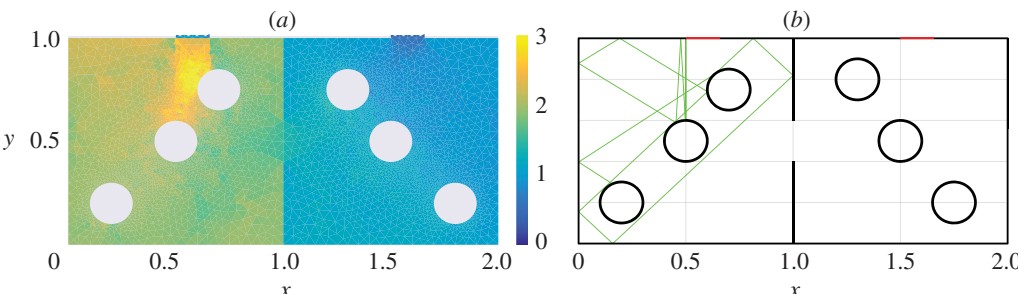

**Figure 4.** Implementation of the cavity shown in figure 1b by the (a) DEA and (b) RT approach. Subplot (a) shows the energy density by means of DEA and (b) shows a typical example of ray trajectory computed by the RT method. In both cases, the incident ray is directed normal from Port 1. (Online version in colour.)

one of the rays tracked by the RT algorithm: The ray bounces within Cavity 1 hitting the scatterers multiple times, and will eventually enter Cavity 2.

For the scatterer positions shown in figures 4a,b, we vary the absorption parameter $\alpha$ (equivalent to the power reflectivity $R = 1 - \alpha$) of both the walls and the scatterers. The calculated powers at Port 1 and Port 2 versus the absorption coefficient $\alpha$ are shown in figure 5 for the three methods. The quantities $P_1^{\text{port}}$ and $P_2^{\text{port}}$ refer to power leaving through Port 1 and Port 2, respectively. As shown in figure 5, DEA and RT follow closely the power balance results in the regime of low and intermediate losses. There is, however, a substantial deviation between DEA and RT on the one hand and PWB on the other hand in the high loss limit. The deviation at high losses can be understood in the following way. Both DEA and RT treat the propagation of energy through the cavities in full. For the given configuration, wave energy entering Port 1 must be

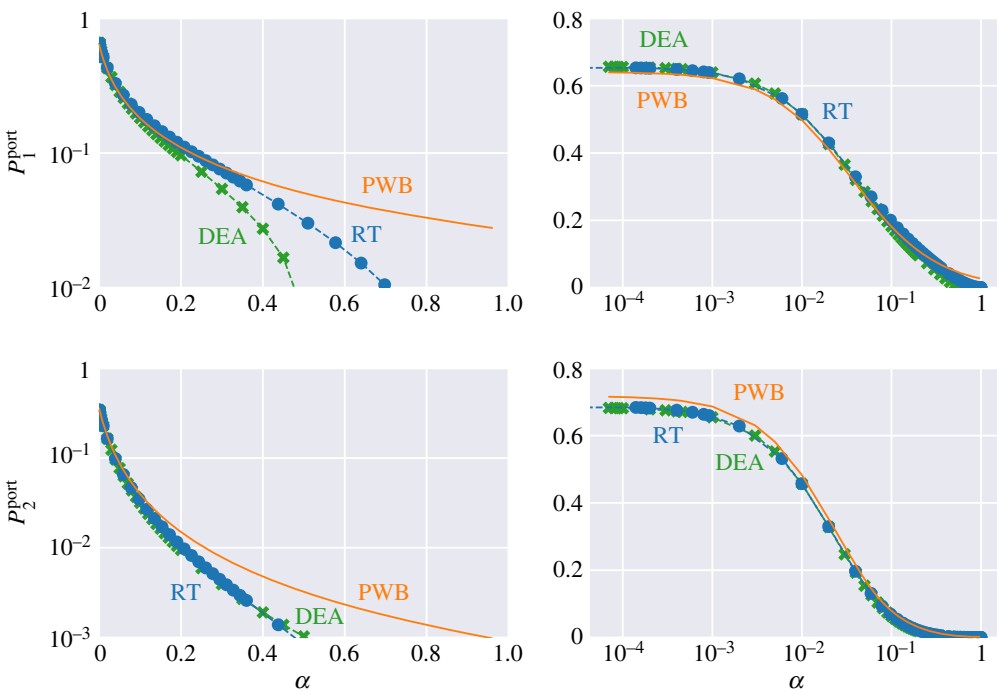

**Figure 5.** The left column shows plots of power on log scale versus loss factor $\alpha$. The right column shows the same data except $\alpha$ is on a log scale and power on a linear scale. $P_1^{\text{port}}$ and $P_2^{\text{port}}$ refer to power escaping through Port 1 and 2, respectively. (Online version in colour.)

reflected by at least one wall or scatterer section before it leaves through Port 1 or Port 2. At high losses, this means a substantially larger fraction of injected power will be lost to the walls than would be predicted based simply on the relative sizes of the ports and the wall. PWB assumes that the power is uniformly distributed within each cavity and thus there is a larger proportion of the energy near ports than in the actual energy flow calculations using RT or DEA. We note that the DEA and RT results match very well for all $\alpha$ except for $P_1^{\text{port}}$ at high loss. This is due to numerical diffusion present in DEA calculations having a larger effect on direct processes, i.e. energy leaving through Port 1 again after one or two reflections. An example of energy diffusion predicted by DEA is presented in [56]. Note also, that in the regime of low losses one finds a small deviation between RT/DEA and PWB. This is due to the fact that different positions of the scatterers lead to slightly different results for DEA/RT, whereas, the exact locations of the scatterers has no bearing on the PWB results. This is illustrated by calculations for the geometry depicted in figure 6a,b where we have moved the scatterers of the left cavity to new positions. This leads to slightly different curves for $P_1^{\text{port}}$ and $P_2^{\text{port}}$ as shown in figure 7.

To explore the origin of the difference between the powers leaving through Port 1 in the two cases, figure 4 and figure 6, we have first simplified the problem by eliminating cavity 2 but keeping the aperture, which now acts similar to Port 1. The geometries we consider are shown in figures 2 and 3. We then inject power through a point source whose location is varied while keeping the scatterers fixed. These results are summarized in figure 8, where we have plotted $P_1^{\text{port}}$ and $P_A^{\text{port}}$ (power leaving through the side aperture) at $\alpha = 0$ versus realizations (that is, the different locations of the point sources). We also show the value based on a power balance estimate alone at $\alpha = 0$, that is,

$$\frac{P_1^{\text{port}}}{P_1^{\text{inj}}} = \frac{w_1}{(w_1 + w_A)} = 0.44.$$

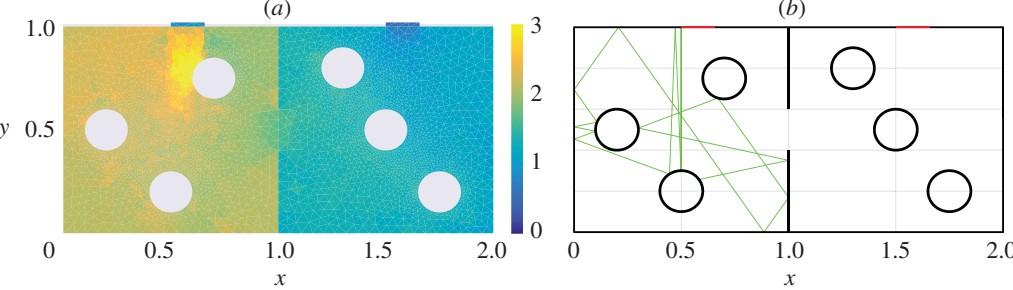

**Figure 6.** We show the implementation of DEA and RT for a cavity for which we moved the scatterers-figure 1a. In particular, (*a*) the computation of wave energy density by means of DEA and (*b*) RT implementation with one sample ray trajectory. Both of these can be compared with the images in figure 4. (Online version in colour.)

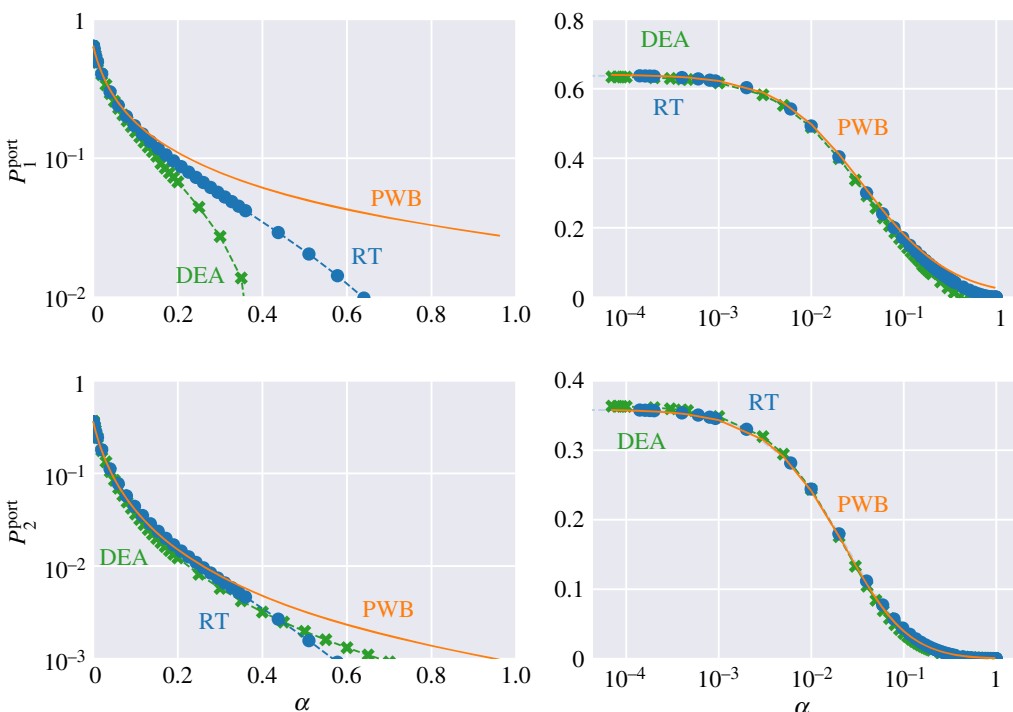

**Figure 7.** Reflected $P_1$ and transmitted $P_2$ EM power as function of absorption parameter $\alpha$ by the three different high-frequency methods. (Left column) power on a log scale versus $\alpha$. (Right column) the same data as the left column displayed in log-linear scale for $\alpha$ and power, respectively. (Online version in colour.)

This relation can be derived from equation (4.6), setting $w_2 \equiv w_A$ and the original aperture width $w_A \to \infty$. This corresponds to no aperture being present and we are in a one-cavity situation with two openings. It can be seen that both DEA and RT values of $P_1^{\text{port}}$ have variations depending on the source position. The RT and DEA results show the same trend overall with the RT results slightly larger than those of DEA. To shed light on these variations, we decrease the size of the apertures and the ports by a factor of 10, see figure 3. These results are summarized in figure 8. This has no effect on the expected value obtained by means of PWB, but it reduces the variations for calculations based on RT and DEA. This can be understood by considering that PWB assumes a uniform distribution of wave energy across the cavities. For RT and DEA cases this holds in

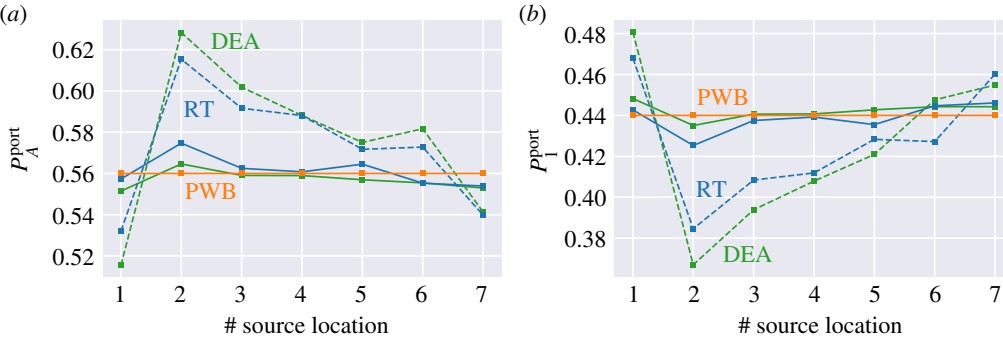

**Figure 8.** Power flux versus point source locations (table 2) for the geometry shown in figures 2 and 3 calculated using the three high-frequency methods at $\alpha = 0$ (no wall damping). Power flux exiting from (*a*) the side Port $P_A^{port}$ and (*b*) the top Port $P_1^{port}$. In (*a,b*), the dotted lines are for the structure in figure 2 and solid lines are for the structure in figure 3. (Online version in colour.)

the limit $\alpha = 0$ and small apertures, that is the wave energy has sufficient time to visit all of the available phase space. In the case shown in figure 2 energy is more likely to escape from the cavity before exploring the whole phase-space and thus leading to the deviation from PWB—ergodicity assumption. This is also apparent in the stronger gradients seen in figure 2 compared to figure 3.

We expect RT and DEA to perform better than PWB in the outdoor regime, where losses are large, i.e. $\alpha > 0.5$ and the ergodic hypothesis is not valid anymore. DEA can be applied to any indoor/outdoor environment whose mesh-based CAD representation is available, and the reflection coefficients of objects is known. DEA does not rely on the ergodic hypothesis (as in PWB) and is computationally efficient for rays surviving tens of thousands of bounces (for which RT becomes computationally demanding) through the environment. Furthermore, either deterministic or stochastics complex sources can be incorporated in DEA through the Wigner function method [57,58], through which the phase of the EM wave field can be accounted for. The classical flux density propagated by DEA is retrieved by the Wigner function by ensemble/frequency averaging. This is important for example in MIMO arrays, where the direction of the beam-steering originates from phased signal driving the antenna array ports: The averaging applied to the Wigner function removes the phase while it captures the direction of propagation of the classical beam through the phase space representation.

In the context of wireless communications, the literature offers some insightful comparison between RT methods and boundary integral equation solvers validated in coupled indoor environments [59]. Given the close agreement between DEA and RT, this may serve as an indirect validation of DEA compared to full wave methods in the context of EM fields. Moreover, DEA has been validated by both FEM simulations [60] and measurements [10] of structure-borne sound transmission. Further research is ongoing to validate RT and DEA predictions for EM wave propagation.

## 6. Conclusion

We have compared under controlled conditions three different approximate methods of computing wireless power distribution in multi-cavity systems. These are the power balance method (PWB), equivalently the statistical energy analysis (SEA), ray tracing (RT) and the dynamic energy analysis (DEA). All three methods apply to situations in which the wavelength of the radiation is much smaller than the typical length scales in the problem. As such these approximate methods are computationally more efficient than full wave computations.

Of the three methods RT and DEA both include propagation effects and account for details of the geometry of the region being modelled. In principle, these two methods are mathematically equivalent, although different in implementation. One can view RT as a Lagrangian description

of wave propagation and DEA as an Eulerian description in analogy with continuum mechanics. The PWB method, however, only follows the total energy in each macroscopic region (cavity) of the system. Energy is assumed to be uniformly distributed throughout that region, and thus propagation effects are not included.

Our findings are as follows. The three methods generally make the same predictions for gross quantities such as the power leaving through ports or dissipated in walls. These are the things that PWB predicts that can be compared directly with corresponding quantities predicted by ray tracing. There are discrepancies between PWB and the other two methods in the prediction of these gross quantities in cases where the assumptions needed for PWB are not met. Specifically, when the energy decay time is too short to allow energy to become uniformly distributed throughout phase space. Examples of this are cases where the wall absorption coefficient is close to unity. A similar situation occurs in configurations where ports are so large that EM wave randomization is incomplete. This leads in the DEA and RT computations to strongly in-homogeneous power distributions that deviate from the average port power predicted by PWB.

In principle, DEA and RT should give identical results in cases where a converged number of rays are followed in RT and where a converged resolution in phase space is used in the DEA. We have found some small deviations between these two methods which we attribute to numerical diffusion in the DEA method, which is due to a limited resolution in the momentum representation. These differences were much smaller than those found by comparing PWB to the two other methods.

It is interesting to ask what further comparisons can be made. First, we might compare the three approximate solutions to full wave solutions to find how the differences among the approximate solutions compare with the difference with the exact solution. Such tests are currently underway. Second, the current study focused on 2D geometry. The issue of three dimensions and the added complication of field polarization should be addressed. Third, the comparisons between DEA and RT were limited to gross quantities. The DEA computes local wave energy density on a grid. Ray Tracing methods can also produce such a quantity. The most efficient way of doing this would be to adopt the particle in cell (PIC) methods of charged particle dynamics. This approach treats the motion of particles in the Lagrangian picture while accumulating on a grid the Eulerian change and current densities. As the next generation of wireless networks are designed and deployed many of these issues will be addressed.

Data accessibility. A bitbucket repository containing results presented here: bitbucket.org/rawData .

Authors' contributions. F.A. implemented and generated results for the RT method, V.B. generated results for the DEA method, S.P. developed the DEA code, G.T. introduced the DEA formulation, T.M.A. developed the PWB formulation, S.C.C. formulated the DEA adjoint approach, S.P., T.M.A., S.C.C., G.G., G.T. supervised the project. All authors analysed the results and wrote the manuscript. All authors reviewed and approved the manuscript.

Competing interests. We declare we have no competing interests.

Funding. Supported by AFOSR/AFRL under FA9550-15-1-0171, ONR under N000141512134 and ONR under N629091612115.

Acknowledgements. We acknowledge the fruitful discussions with Martin Richter (University of Nottingham), Edward Ott and Stephen Anlage (University of Maryland).

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
