## [Reviewer comments · Proceedings. Mathematical, Physical, and Engineering Sciences]

Review History

RSPA-2020-0228.R0 (Original submission)

Review form: Referee 1

Is the manuscript an original and important contribution to its field?

Good

Is the paper of sufficient general interest?

Acceptable

Is the overall quality of the paper suitable?

Acceptable

Can the paper be shortened without overall detriment to the main message?

Yes

Do you think some of the material would be more appropriate as an electronic appendix?

No

Do you have any ethical concerns with this paper?

No

Recommendation?

Accept with minor revision (please list in comments)

Comments to the Author(s)

There are new ideas in the design scheme with theoretical novelties

Review form: Referee 2

Is the manuscript an original and important contribution to its field?

Excellent

Is the paper of sufficient general interest?

Excellent

Is the overall quality of the paper suitable?

Good

Can the paper be shortened without overall detriment to the main message?

Yes

Do you think some of the material would be more appropriate as an electronic appendix?

No

Do you have any ethical concerns with this paper?

No

Recommendation?

Accept with minor revision (please list in comments)

Comments to the Author(s)

The paper is well written and certainly of interest for the scientific community. Consequently, I recommend its publication in the journal, after some minor review will be completed by the authors. Some points the authors should go through in their review are listed in the following.

1. I suggest to add a reference to Table 1 at line 34 of page 10 of the manuscript, changing: "The points labelled 1 ... 7 indicate the locations of sources" into "The points labelled 1 ... 7 indicate the locations of sources reported in Table 1,"
2. It would be better to give reference to Table 2 in the text, which is not present now
3. At the end of Section 2b (Ray tracing), the authors say that they do not consider in their ray tracing simulation ray lengths, so that interferences are not included. The authors should explain how this is affecting the accuracy of ray tracing calculations. Also, when presenting Section 5. Results and Discussions, there is no indication about how many reflections from obstacles are taken into account in ray tracing calculations. Finally, for the kind of obstacles chosen, it is apparent that diffraction has not any influence in the results shown. For future work it would be interesting if some different geometry, including geometrical or electrical discontinuities could be considered in the environment.

Review form: Referee 3

Is the manuscript an original and important contribution to its field?

Acceptable

Is the paper of sufficient general interest?

Acceptable

Is the overall quality of the paper suitable?

Good

Do you have any ethical concerns with this paper?

No

Recommendation?

Accept with minor revision (please list in comments)

Comments to the Author(s)

The research and ideas presented in the manuscript are well written and easily readable. The comparisons presented are useful, it would be interesting to see how the RT/DEA compares against full wave solutions.

Some minor corrections to improve the readability are suggested.

Decision letter (RSPA-2020-0228.R0)

29-Oct-2020

Dear Colleagues,

I am writing to inform you that the Editor has made a decision on manuscript entitled "Wireless Power Distributions in Multi-Cavity Systems at High Frequencies" which you kindly refereed for Proceedings A. Please find the authors' decision letter below.

On behalf of the Editor of Proceedings A, we thank you for your help with this article and we look forward to your input in the future.

Decision made on this manuscript: Accept with minor revision

Best wishes

Raminder Shergill

proceedingsa@royalsociety.org

29-Oct-2020

Dear Dr Gradoni,

On behalf of the Editor, I am pleased to inform you that your Manuscript RSPA-2020-0228 entitled "Wireless Power Distributions in Multi-Cavity Systems at High Frequencies" has been accepted for publication subject to minor revisions in Proceedings A. Please find the referees' comments below.

The reviewer(s) have recommended publication, but also suggest some minor revisions to your manuscript. Therefore, I invite you to respond to the reviewer(s)' comments and revise your manuscript. Please note that we have a strict upper limit of 28 pages for each paper. Please endeavour to incorporate any revisions while keeping the paper within journal limits. Please note that page charges are made on all papers longer than 20 pages. If you cannot pay these charges you must reduce your paper to 20 pages before submitting your revision. Your paper has

been ESTIMATED to be 16 pages. We cannot proceed with typesetting your paper without your agreement to meet page charges in full should the paper exceed 20 pages when typeset. If you have any questions, please do get in touch.

It is a condition of publication that you submit the revised version of your manuscript within 7 days. If you do not think you will be able to meet this date please let me know in advance of the due date.

To revise your manuscript, log into <https://mc.manuscriptcentral.com/prsa> and enter your Author Centre, where you will find your manuscript title listed under "Manuscripts with Decisions." Under "Actions," click on "Create a Revision." Your manuscript number has been appended to denote a revision.

You will be unable to make your revisions on the originally submitted version of the manuscript. Instead, revise your manuscript and upload a new version through your Author Centre.

IMPORTANT: Your original files are available to you when you upload your revised manuscript. Please delete any redundant files before completing the submission process.

In addition to addressing all of the reviewers' and editor's comments, your revised manuscript **MUST** contain the following sections before the reference list (for any heading that does not apply to your work, please include a comment to this effect):

- Acknowledgements
- Funding statement

See <https://royalsociety.org/journals/authors/author-guidelines/> for further details.

When uploading your revised files, please make sure that you include the following as we cannot proceed without these:

- 1) A text file of the manuscript (doc, txt, rtf or tex), including the references, tables (including captions) and figure captions. Please remove any tracked changes from the text before submission. PDF files are not an accepted format for the "Main Document".
- 2) A separate electronic file of each figure (tif, eps or print-quality pdf preferred). The format should be produced directly from original creation package, or original software format.
- 3) Electronic Supplementary Material (ESM): all supplementary materials accompanying an accepted article will be treated as in their final form. Note that the Royal Society will not edit or typeset supplementary material and it will be hosted as provided. Please ensure that the supplementary material includes the paper details where possible (authors, article title, journal name). Supplementary files will be published alongside the paper on the journal website and posted on the online figshare repository (<https://figshare.com>). The heading and legend provided for each supplementary file during the submission process will be used to create the figshare page, so please ensure these are accurate and informative so that your files can be found in searches. Files on figshare will be made available approximately one week before the accompanying article so that the supplementary material can be attributed a unique DOI.

Alternatively you may upload a zip folder containing all source files for your manuscript as described above with a PDF as your "Main Document". This should be the full paper as it appears when compiled from the individual files supplied in the zip folder.

Article Funder

Please ensure you fill in the Article Funder question on page 2 to ensure the correct data is collected for FundRef (<http://www.crossref.org/fundref/>).

Media summary

Please ensure you include a short non-technical summary (up to 100 words) of the key findings/importance of your paper. This will be used for to promote your work and marketing purposes (e.g. press releases). The summary should be prepared using the following guidelines:

*Write simple English: this is intended for the general public. Please explain any essential technical terms in a short and simple manner.

*Describe (a) the study (b) its key findings and (c) its implications.

*State why this work is newsworthy, be concise and do not overstate (true 'breakthroughs' are a rarity).

*Ensure that you include valid contact details for the lead author (institutional address, email address, telephone number).

Cover images

We welcome submissions of images for possible use on the cover of Proceedings A. Images should be square in dimension and please ensure that you obtain all relevant copyright permissions before submitting the image to us. If you would like to submit an image for consideration please send your image to proceedingsa@royalsociety.org

Once again, thank you for submitting your manuscript to Proceedings A and I look forward to receiving your revision. If you have any questions at all, please do not hesitate to get in touch.

Best wishes

Raminder Shergill

proceedingsa@royalsociety.org

Proceedings A

on behalf of

Dr Anas Al Rawi

Board Member

Proceedings A

Reviewer(s)' Comments to Author:

Referee: 1

Comments to the Author(s)

There are new ideas in the design scheme with theoretical novelties

Referee: 2

Comments to the Author(s)

The paper is well written and certainly of interest for the scientific community. Consequently, I recommend its publication in the journal, after some minor review will be completed by the authors. Some points the authors should go through in their review are listed in the following.

1. I suggest to add a reference to Table 1 at line 34 of page 10 of the manuscript, changing: "The points labelled 1 ... 7 indicate the locations of sources" into "The points labelled 1 ... 7 indicate the locations of sources reported in Table 1,"
2. It would be better to give reference to Table 2 in the text, which is not present now
3. At the end of Section 2b (Ray tracing), the authors say that they do not consider in their ray tracing simulation ray lengths, so that interferences are not included. The authors should explain how this is affecting the accuracy of ray tracing calculations. Also, when presenting Section 5. Results and Discussions, there is no indication about how many reflections from obstacles are taken into account in ray tracing calculations. Finally, for the kind of obstacles chosen, it is apparent that diffraction has not any influence in the results shown. For future work it would be interesting if some different geometry, including geometrical or electrical discontinuities could be considered in the environment.

Referee: 3

Comments to the Author(s)

The research and ideas presented in the manuscript are well written and easily readable. The comparisons presented are useful, it would be interesting to see how the RT/DEA compares against full wave solutions.

Some minor corrections to improve the readability are suggested.

Board Member:

Comments to Author(s):

The authors in this paper discussed the challenges facing the wireless community concerning the prediction of propagation conditions of emerging wireless technologies. In particular, millimetre-wave frequencies lack well-established and tested statistical models based on extensive measurement campaigns to aid technology development. Numerical solutions are computationally expensive and inversely proportional to the wavelength as the authors pointed out. To address the present challenge, the authors compared three different methods for calculating power distribution of wireless systems in closed cavities. These are the Power Balance Method, Ray Tracing (RT), and the Dynamic Energy Analysis (DEA). The paper is well written, easy to follow and provides and discusses a rich list of references. In addition to the reviewers comments, below I provide a list of additional comments to the authors to address:

1- I echo one of the reviewers' comments regarding model validation, how does these solutions compare to a full wave solution? please discuss and share your insight or findings. Perhaps this is an activity worth pursuing in your future research to aim at validating your models against full wave solutions and/or experimental tests.

2- In the introduction the authors suggest that 5G is a mmwave-based technology. This is partially true, in the UK and Europe 3.6GHz carrier frequency is adopted whilst mmwave in the USA and Asia.

3- Page 6, line 49, typo in "to chose" 4- Section 4, please assign numbers to all equations unless there is a specific reason to skip the first 4.

5- Section 4, page 9, line 42, σ_{wall} seems to have a unit of length (metre?), please explain this. Also, is it linear or logarithmic? it seems linear with α , please elaborate on this. Same comment applies to σ_{tot} .

6- Shouldn't there be an integration involved in P_{tot} and σ_{tot} since the paper discusses 2D power distribution?

7- The RT technique is not discussed in section 4, please provide a reference or subsection on the model.

8- The results of figures 6 and 7 seem to extend beyond the figures' frames

9- The models focused on a single source per cavity, however modern communication systems rely on using phased array MIMO, please discuss potential scalability to MIMO and insight/techniques to expand the models to cover phase modelling as well as outdoor environments.

Overall, the paper has indeed picked on a fundamental challenge and of great importance to dimension future network coverage and performance. Whilst this paper reviews the literature extensively and provides some answers, the authors are encouraged to invest more efforts in this subject and attempt to translate their knowledge/models more effectively to the wireless community in a much more conventional format (e.g. s parameters, path loss, interference, MIMO channel... etc) to facilitate applying these models to aid and accelerate next generation technology development.

Author's Response to Decision Letter for (RSPA-2020-0228.R0)

See Appendix A.

Decision letter (RSPA-2020-0228.R1)

25-Nov-2020

Dear Dr Gradoni

I am pleased to inform you that your manuscript entitled "Wireless Power Distributions in Multi-Cavity Systems at High Frequencies" has been accepted in its final form for publication in Proceedings A.

Our Production Office will be in contact with you in due course. You can expect to receive a proof of your article soon. Please contact the office to let us know if you are likely to be away from e-mail in the near future. If you do not notify us and comments are not received within 5 days of sending the proof, we may publish the paper as it stands.

Open access

You are invited to opt for open access, our author pays publishing model. Payment of open access fees will enable your article to be made freely available via the Royal Society website as soon as it is ready for publication. For more information about open access please visit <https://royalsociety.org/journals/authors/which-journal/open-access/>. The open access fee for this journal is £1700/\$2380/€2040 per article. VAT will be charged where applicable.

Note that if you have opted for open access then payment will be required before the article is published – payment instructions will follow shortly.

If you wish to opt for open access then please inform the editorial office (proceedingsa@royalsociety.org) as soon as possible.

Under the terms of our licence to publish you may post the author generated postprint (ie. your accepted version not the final typeset version) of your manuscript at any time and this can be made freely available. Postprints can be deposited on a personal or institutional website, or a

recognised server/repository. Please note however, that the reporting of postprints is subject to a media embargo, and that the status the manuscript should be made clear. Upon publication of the definitive version on the publisher's site, full details and a link should be added.

You can cite the article in advance of publication using its DOI. The DOI will take the form: 10.1098/rspa.XXXX.YYYY, where XXXX and YYYY are the last 8 digits of your manuscript number (eg. if your manuscript number is RSPA-2017-1234 the DOI would be 10.1098/rspa.2017.1234).

For tips on promoting your accepted paper see our blog post:
<https://royalsociety.org/blog/2020/07/promoting-your-latest-paper-and-tracking-your-results/>

On behalf of the Editor of Proceedings A, we look forward to your continued contributions to the Journal.

Sincerely,
Raminder Shergill
proceedingsa@royalsociety.org

on behalf of
Dr Anas Al Rawi
Guest Editor
Proceedings A

Comments to Author(s)

I thank the authors for their patience with the review process and for satisfying the reviewers' comments.

Appendix A

Manuscript RSPA-2020-0228 entitled "Wireless Power Distributions in Multi-Cavity Systems at High Frequencies"

Authors: Farasatul Adnan, Valon Blakaj, Sendy Phang, Thomas M. Antonsen, Stephen C. Creagh, Gabriele Gradoni, and Gregor Tanner

University of Maryland and University of Nottingham.

[Author text in blue.]

We are grateful to the three referees and the Board Member for their detailed reviews and insightful comments. We are pleased that the paper has been accepted for publication following some minor corrections and we have now completed a careful review of the manuscript.

Referee: 1

Comments to the Author(s)

There are new ideas in the design scheme with theoretical novelties

Answer to the Referee

We thank the Referee for the positive feedback.

Referee: 2

Comments to the Author(s)

The paper is well written and certainly of interest for the scientific community. Consequently, I recommend its publication in the journal, after some minor review will be completed by the authors.

Answer to the Referee

We thank the Referee for the points for improvement suggested below and for recommending the publication of our paper.

Some points the authors should go through in their review are listed in the following.

1. I suggest to add a reference to Table 1 at line 34 of page 10 of the manuscript, changing: "The points labelled 1 ... 7 indicate the locations of sources" into "The points labelled 1 ... 7 indicate the locations of sources reported in Table 1,"

Answer to the Referee

Thanks, we have just amended the sentence and added a reference to Table 1 as per your suggestion.

2. It would be better to give reference to Table 2 in the text, which is not present now

Answer to the Referee

We are grateful to the Referee for noting the missing reference to Table 2. This has now been added by including the sentence: “Table 2 reports the dimensions of port 1 (P_1) in cavity 1, port 2 (P_2) in cavity 2, and the aperture (P_A) in the wall connecting the two cavities.” at the end of Section 3.

3. At the end of Section 2b (Ray tracing), the authors say that they do not consider in their ray tracing simulation ray lengths, so that interferences are not included. The authors should explain how this is affecting the accuracy of ray tracing calculations.

Answer to the Referee

This is a good point, we thank you for your insightful comment. Since DEA is based on energy flow equations, the method predicts the average energy/power distribution within a given system. In this context, wave-field phases disappear under averaging (the so-called diagonal approximation) and interference is not included in the formulation. In order to perform a meaningful comparison with DEA, the RT formulation has also been constructed on power-based arguments (see answer to comment 7- of *Board Member*). This can be interpreted as predicting the average Path Loss within the coupled-cavity environment: Interference can be captured separately by a statistical distribution describing large/small scale fading. An example of this procedure has been given in [Ma *et al* Phys. Rev. Applied 14, 014022 – Published 8 July 2020 - arxiv.org/abs/2003.07942] combining the PWB and the Random Coupling Model. A different procedure is presented in [Creagh *et al* 2016 J. Phys. A: Math. Theor. 50 045101] where it is shown how to estimate the variance of the energy from DEA predictions. The average energy predicted by RT and DEA are accurate in the moderate to high loss regime, which is a reasonable assumption in wireless environments.

The following sentence has been added at the end of Section 2. (c): “We note that neither PWB nor DEA carry any phase information; the results obtained give average energy densities and do not try to replicate wave fluctuations due to interference and resonance phenomena. *This mirrors the diagonal approximation adopted to obtain the DEA flow equations, where the phase difference between waves disappear under averaging. Interference induced fluctuations between waves can be predicted separately by statistical methods to give a probability distribution around the average energy provided by DEA/RT discussed in this paper (see [Ma *et al*] for an example involving coupled cavities). Furthermore, it has been shown in [Creagh *et al*] that the energy variance can be estimated via the autocorrelation of DEA densities. RT and DEA predictions are accurate in the moderate to high loss regime, which is a reasonable assumption in both indoor and outdoor wireless communication environments.*”

Also, when presenting Section 5. Results and Discussions, there is no indication about how many reflections from obstacles are taken into account in ray tracing calculations. Finally, for the kind of obstacles chosen, it is apparent that diffraction has not any influence in the results shown. For future work it would be interesting if some different geometry, including geometrical or electrical discontinuities could be considered in the environment.

Answer to the Referee

Those are important points, thanks. We have amended a sentence in Section 5. to include both the number of rays and bounces used in the RT algorithm “*In the corresponding RT*

calculation a total of 8002 rays were launched normal to the boundary of Port 1, and each ray is followed for 25 bounces off the walls and object boundaries.” and a reference to our recent work [Creagh et al 2020 J. Phys. A: Math. Theor., in press] on including the geometrical theory of diffraction in Eulerian ray tracing methods has been added in the introduction. This work is the first step towards including diffraction from objects in DEA: “Recently, a theory has been developed to include the geometrical theory of diffraction in phase space distributions through the Wigner function [Creagh et al 2020] which paves the way to the inclusion of diffraction in DEA.”

Referee: 3

Comments to the Author(s)

The research and ideas presented in the manuscript are well written and easily readable. The comparisons presented are useful, it would be interesting to see how the RT/DEA compares against full wave solutions.

Answer to Referee 3 – see also answer to the Board Member below

In currently ongoing research in a separate in-depth study we are adopting a commercial software to computer the full wave solution for a coupled cavity scenario at various frequency bands. While we take the point made by the *Referee* and the *Board Member* (below) that a comparison with full wave calculations could have been done already in this paper, we prefer to limit the study here to a comparison between the flow methods . Follow on work also considering comparison between DEA, RT with phase information, full wave calculations and DEA/RCM hybrid methods (extending the results of MA et al 2020) is in preparation. A few sentences discussing the present literature have been added in Section 5. *“In the context of wireless communications, the literature offers some insightful comparison between RT methods and boundary integral equation solvers validated in coupled indoor environments [Kavanagh et al 2016]. Given the close agreement between DEA and RT, this may serve as an indirect validation of DEA compared to full wave methods in the context of electromagnetic fields. Moreover, DEA has been validated by both finite element method (FEM) simulations [Chappell et al 2012] and measurements [Hartmann et al 2019] of structure-borne sound transmission. Further research is ongoing to validate RT and DEA predictions for electromagnetic wave propagation.”*

Some minor corrections to improve the readability are suggested.

Answer to the Referee

We thank the Reviewer for the detailed comments listed in the pdf file. We have now addressed all the comments in full, including an expression defining the inner product in (4.9).

Board Member:

Comments to Author(s):

The authors in this paper discussed the challenges facing the wireless community concerning the prediction of propagation conditions of emerging wireless technologies. In particular, millimetre-wave frequencies lack well-established and tested statistical models

based on extensive measurement campaigns to aid technology development. Numerical solutions are computationally expensive and inversely proportional to the wavelength as the authors pointed out. To address the present challenge, the authors compared three different methods for calculating power distribution of wireless systems in closed cavities. These are the Power Balance Method, Ray Tracing (RT), and the Dynamic Energy Analysis (DEA). The paper is well written, easy to follow and provides and discusses a rich list of references.

Answer to the Board Member

We thank the Board Member for valuing our work.

In addition to the reviewers comments, below I provide a list of additional comments to the authors to address:

- 1- I echo one of the reviewers' comments regarding model validation, how does these solutions compare to a full wave solution? please discuss and share your insight or findings. Perhaps this is an activity worth pursuing in your future research to aim at validating your models against full wave solutions and/or experimental tests.

Answer to the Board Member

We thank the Board Member, and again Referee 3, for touching on this interesting aspect. While the three methods have been validated individually with full wave simulations and measurements, an in-depth study is ongoing to validate the coupled room scenario presented in this paper. Please see also our reply to referee 3 above.

We have added a few sentences regarding the literature in this area at the end of Section 5 (see answer to the comments of Referee 3).

2- In the introduction the authors suggest that 5G is a mmwave-based technology. This is partially true, in the UK and Europe 3.6GHz carrier frequency is adopted whilst mmwave in the USA and Asia.

Answer to the Board Member

We agree that the information in the Introduction are outdated. We have now amended the sentences around that part: "*The next generation of wireless communication systems (5G) will be based on Super High Frequency (SHF) and Extremely High Frequency (EHF) technologies making use of the enormous amount of bandwidth available at high frequency bands [Rappaport et al 2013].*

Currently, the United Kingdom and Europe have adopted carrier frequencies around 3.6 GHz. The Americas and Asia will operate at both sub-6 GHz bands as well as mmwave bands: more precisely, carrier frequencies at 24 GHz and 28 GHz have been adopted [Xiao et al 2017, Rappaport et al 2017]."

3- Page 6, line 49, typo in "to chose" 4- Section 4, please assign numbers to all equations unless there is a specific reason to skip the first 4.

Answer to the Board Member

We are grateful to the Board Member for the suggestions. We have now fixed the typo and labelled the first four equations in Section 4.

5- Section 4, page 9, line 42, σ_{i}^{wall} seems to have a unit of length (metre?), please explain this. Also, is it linear or logarithmic? it seems linear with α , please elaborate on this. Same comment applies to σ_{i}^{tot} .

Answer to the Board Member

We agree with the interpretation of the Board Member concerning the quantities mentioned. First, α is the fraction of (incident) power absorbed locally by the wall. It is not measured logarithmically in dB, but rather linearly. Second, regarding the units, we are considering a 2D situation, which can make keeping track of quantities' units confusing. Normally, in a 3D enclosure the power absorbed by an object is determined by the product of its area and the energy per unit volume in the enclosure. The proportionality constant thus has units of m/s reflecting the speed of wave propagation. In our 2D world the power absorbed by an object is proportional to the length of the object and the energy per unit area. Again, the constant of proportionality has units m/s. If our 2D object is considered to be the cross section of a 3D object that is extended uniformly a distance L in the third direction then the energy per unit area is L times the energy per unit volume of the 3D object, and the area of the absorber is L times its length in the 2D plane. The L 's cancel.

We have added the following after the first appearance of a σ :

“In our 2D system the absorption cross-section represents an effective physical length. The power absorbed is proportional to the product of this length and the energy per unit area in the cavity under consideration. The constant of proportionality scales with the wave speed. In addition, the power lost by escaping through the cavity walls is proportional to the fraction of power incident α , which in practice is determined by the electrical characteristics of the wall material. Here, we have not focused on specific/prescribed wall materials: We rather performed a study by varying α across the full range, from 0 (the incident power is entirely reflected) to 1 (the incident power is entirely absorbed).

...

Similarly, to σ_{i}^{wall} , σ_{i}^{tot} in (4.2) represents the total length through which the power escapes the cavity.”

6- Shouldn't there be an integration involved in P_{i}^{tot} and σ_{i}^{tot} since the paper discusses 2D power distribution?

Answer to the Board Member

We thank the Board Member for the insightful question. This is correct in principle, and it is done in RT/DEA. However, in the power balance method, the ergodicity assumption allows for treating the average energy within the cavity as constant, i.e., independent on the spatial location. Therefore, integration produces lengths and calculations can be performed by application of energy conservation.

7- The RT technique is not discussed in section 4, please provide a reference or subsection on the model.

Answer to the Board Member

We thank the Board Member for identifying this gap. We have now added a new section, 4 c, which is also reported below.

“4. Calculation of power transfer

(b) Power transfer in the RT method

We briefly discuss the RT algorithm used to calculate power delivery in this section. We basically follow one of the approaches discussed in [Z. Yun et al 2015]. Specifically, we use the 'Reflected and Transmitted Rays' approach as discussed in Section III B and the power calculation is done using the 'Shooting and Bouncing Ray (SBR) Method' as discussed in Section IV C of [Z. Yun et al 2015].

At a port we launch rays normal to the boundary and uniformly distributed over the port. Each ray contains initially the same amount of power. The rays follow straight trajectories until they encounter a wall or scatterer at which point they are specularly reflected and their power is reduced according to

$$P_{(n+1)} = |R|^2 P_n ,$$

where n refers to the bounce number, P_n is the power contained in the ray after the n-th bounce and R is the power reflectivity (discussed further in the next section). So, after each bounce the ray loses (P_(n) - P_(n+1)) amount of power. The R in this paper is independent of incident angle of the rays and we can fix a value of R for a particular run. Rays that encounter the aperture are allowed to pass straight through, and rays that encounter a port are allowed to leave. We add up all the ray powers when the rays escape through a port and that gives us the total power delivered to a port. To monitor the energy in each cavity we sum the contributions from each ray segment in a cavity, taking the product of the power and the time of flight of the segment.

Numerical details are as follows. We typically launch 8002 rays from the port and we let the rays bounce until they escape through either port 1 or port 2 (as shown in Fig. 4(b)). We have found 20 bounces to be enough for all rays to escape.” [NOTE: the ray number discussion is done in Section 5 of our paper also]. The RT code used here was developed in MATLAB [<https://www.mathworks.com/products/matlab.html>].

8- The results of figures 6 and 7 seem to extend beyond the figures' frames

Answer to the Board Member

We thank the Board Member for point out this potential problem. Although we do not seem to get bounding box problems, we have reduced the width of the two figures to 0.8\textwidth.

9- The models focused on a single source per cavity, however modern communication systems rely on using phased array MIMO, please discuss potential scalability to MIMO and insight/techniques to expand the models to cover phase modelling as well as outdoor environments.

Answer to the Board Member

We thank the Board Member for the interesting comment. We have added a few sentences at the end of Section 5. that address MIMO antenna arrays and outdoor modelling.

“We expect RT and DEA to perform better than PWB in the outdoor regime, where losses are large, i.e., $\alpha > 0.5$ and the ergodic hypothesis is not valid anymore. DEA can be applied to any indoor/outdoor environment whose mesh-based CAD representation is available, and the reflection coefficients of objects is known. DEA does not rely on the ergodic hypothesis (as in PWB) and is computationally efficient for rays surviving tens of thousands of bounces (for which RT becomes computationally demanding) through the environment. Furthermore, either deterministic or stochastic complex sources can be incorporated in DEA through the Wigner function method [Gradoni et al 2018], through which the phase of the electromagnetic wave field can be accounted for. The classical flux density propagated by DEA is retrieved by the Wigner function by ensemble/frequency averaging. This is important for example in MIMO arrays, where the direction of the beam-steering originates from phased signal driving the antenna array ports: The averaging applied to the Wigner function removes the phase while it captures the direction of propagation of the classical beam through the phase space representation.”

Overall, the paper has indeed picked on a fundamental challenge and of great importance to dimension future network coverage and performance. Whilst this paper reviews the literature extensively and provides some answers, the authors are encouraged to invest more efforts in this subject and attempt to translate their knowledge/models more effectively to the wireless community in a much more conventional format (e.g. s parameters, path loss, interference, MIMO channel... etc) to facilitate applying these models to aid and accelerate next generation technology development.

Answer to the Board Member

We appreciate the positive feedback of the Board Member and the Referees and we agree with the last comment of the Board Member. We find it important to develop the work further towards the wireless communication community. There is an ongoing effort to tackle part of this challenge and bridge the gap between physics-based and communication channel models in the context of MIMO and RIS-assisted massive MIMO systems. We would like to stress that the results presented in this paper provide predictions of radio maps, e.g., RSSI levels, which can be used right away in the planning of wireless coverage as well as in the calibration of localisation algorithms. An extension of the formalism that relates directly to path loss, signal-to-interference-plus-noise ratio (in presence of an additional sources which interferes with the information signal), and average channel gains in MIMO systems is achievable in the near future.